# Array Pattern Synthesis Using a Hybrid Differential Evolution and Analytic Algorithm

Rui Li [1], Le Xu [1,*], Xiaoqun Chen [2], Yong Yang [3], Xiaoning Yang [3], Jianxiao Wang [4], Yuanming Cai [1] and Feng Wei [1]

1   Key Laboratory of Antennas and Microwave Technology, Xidian University, Xi'an 710071, China; ruili@mail.xidian.edu.cn (R.L.); ymcai@xidian.edu.cn (Y.C.); fwei@mail.xidian.edu.cn (F.W.)
2   Institute of Telecommunication Satellite, CAST, Beijing 100094, China; cxq_98@126.com
3   Beijing Institute of Spacecraft Environment Engineering, Beijing 100089, China; luojiayang@outlook.com (Y.Y.); yangxiaoning_1@outlook.com (X.Y.)
4   National Key Laboratory of Science and Technology on Space Microwave, China Academy of Space Technology, Xi'an 710100, China; wjx8902@126.com
*   Correspondence: lexu@mail.xidian.edu.cn

**Abstract:** In this paper, a hybrid differential evolution and weight total least squares method (HDE-WTLSM) is proposed for antenna array pattern synthesis. A variable diagonal weight matrix is introduced in total least squares method. Then, the weight matrix is optimized by differential evolution (DE) algorithm to control the differences of the desired level and the obtained level in different directions. This algorithm combines the advantages of evolutionary algorithm and numerical algorithm, so it has a wider application range and faster convergence speed. To compare HDE-WTLSM with DE algorithm and typical numerical algorithms, these methods are applied to a linear antenna array and a conformal truncated conical array. Using our method, lower sidelobe levels and deeper nulls are obtained. The simulation results verify the validity and efficiently of HDE-WTLSM.

**Keywords:** antenna pattern synthesis; differential evolution (DE); total least squares method (TLSM)

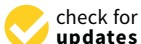



## 1. Introduction

Due to the applications in radar and communication systems, antenna array pattern synthesis has attracted attention over the last several decades. Among them, design of conformal antenna arrays are great challenges to the researchers [1–4]. As the conformal arrays have curved surfaces, each antenna in the array direct in a different direction. Thus, classical array theory for planar arrays fails to provide typical method to synthesis the array radiation pattern. In addition, sometimes null-forming is necessary in the radiation pattern of antenna array in a receiver system. The null positions are controlled to point to the directions of the interference signal to achieve interference suppression. The level of nulling is much lower than that of sidelobe and main lobe, which will bring difficulties to the optimization algorithm. In the field of antenna array synthesis, many analytical methods have been developed, which can realize typical array synthesis, such as Chebyshev, Fourier transform and so on. The analytic methods have prominent advantages in convergence speed, but their application are limited. Least squares method (LSM) is widely used not only in traditional measurement adjustment, but also in modern adjustment theories such as least square fitting and least square collocation. It has been widely used not only in the field of surveying and mapping, but also in many other fields of science and engineering technology. Algorithms based on the LSM are presented in [5–8]. This kind of algorithm is solved in an iterative way and good results can be obtained for linear array synthesis. As conformal array synthesis is faced with the difference of array element attitude, it is difficult to apply the analytical method directly, which limits the development of this kind

of algorithm. The direct method is used in WTLSM to solve array synthesis problem, but it is not suitable to deal with conformal array synthesis problem owing to the constant weight matrix [9].

In the development of conformal array design technology, many scholars have developed conformal array synthesis algorithm based on bionic optimization algorithm. For example, genetic algorithm (GA), particle swarm optimization (PSO), differential evolution (DE) algorithm and so on. Among them, DE algorithm is proposed in [10] and it has been utilized to solve many different kinds of optimization problems, such as subarrayed linear antenna arrays design [11], axis-parallel decision trees [12], multimodal optimization problem [13] and to search the optimal location of battery-swapping stations in a certain region [14].

As DE algorithm is simple in principle and the control parameters are less, it is easy to be understand and realized. It can deal with complex function optimization, global optimization and multi-objective optimization problems effectively. Although DE is excellent in maintain the diversity of population and global searching ability, the convergence speed is slow for some complex problems, such as the pattern synthesis for conformal antenna arrays. To synthesize complex arrays, various types of optimization methods are developed. An improved NSGA-II algorithm is introduced to optimize a conformal phased array in [15]. A blind beamforming method based on linearly constrained minimum variance algorithm is proposed in [16]. However, the convergence speeds of some bionic optimization algorithm are slow, which restrict the application of these algorithms.

In this paper, a hybrid differential evolution and weight total least squares method (HDE-WTLSM) is proposed for antenna arrays to control the sidelobe level (SLL), the main lobe and the null steering (beamwidth and direction). The method can be used to control the main lobe and the SLL in the co-polarization and the cross-polarization level simultaneously. DE is applied to optimize the weight matrix is in WTLSM. Evolutionary algorithm and numerical algorithm are combined, so they have the advantages of wider applicability and faster convergence speed. The embedded patterns of antennas in the array are used to improve the accuracy of pattern synthesis.

This paper is organized as follows: the principle of WTLSM and DE is illustrated in Section 2. Section 3 contains several antenna arrays examples synthesized using HDE-WTLSM. The simulation of low SLLs and null steering synthesis of a linear array are given. Then, the synthesis results of a conformal truncated conical array with low SLLs and low cross-polarization are displayed in this section. The work is concluded in Section 4.

## 2. Hybrid Differential Evolution and Total Least Squares Method

### 2.1. Weighted Total Least Squares Method

Array pattern synthesis problem can be written as follows:

$$[\mathbf{A}][\mathbf{I}] = [\mathbf{S}] \tag{1}$$

where matrix **A** is the array manifold.

$$
[\mathbf{A}]_{M \times N} = [a_{ij}]_{M \times N} =
$$
$$
\begin{bmatrix}
f_1(\theta_1) & \cdots & f_n(\theta_1)e^{jkd_n \cos \theta_1} & \cdots & f_N(\theta_1)e^{jkd_N \cos \theta_1} \\
\vdots & & \vdots & & \vdots \\
f_1(\theta_m) & \cdots & f_n(\theta_m)e^{jkd_n \cos \theta_m} & \cdots & f_N(\theta_m)e^{jkd_N \cos \theta_m} \\
\vdots & & \vdots & & \vdots \\
f_1(\theta_M) & \cdots & f_n(\theta_M)e^{jkd_n \cos \theta_M} & \cdots & f_N(\theta_M)e^{jkd_N \cos \theta_M}
\end{bmatrix} \tag{2}
$$

$$a_{i,j} = f_j(\theta_i)e^{jkd_j \cos \theta_i}, (i = 1, \ldots, M; j = 1, \ldots, N) \tag{3}$$

$$k = \frac{2\pi}{\lambda} \tag{4}$$

$\lambda$ is the wavelength. $M$ is sample points number. $N$ is the total cells number in an array and $n$ stands for the $n$th cell. $dn$ is the distance between the $n$th cell and the first cell.

**I** are the currents of cells.

$$[\mathbf{I}]_{N\times 1} = \left[ I_1 e^{j\phi_1}, \ldots, I_j e^{j\phi_j}, \ldots, I_N e^{j\phi_N} \right]^T \tag{5}$$

$I_n$ ($n = 1, \ldots, N$) is the current of the $n$th cell.

The desired radiation pattern for the array is expressed as vector **S**.

$$[\mathbf{S}]_{1\times M} = [S(\theta_1), \ldots, S(\theta_i), \ldots, S(\theta_M)]^T \tag{6}$$

$$S(\theta_i) = \sum_{n=1}^{N} I_n f_n(\theta_i) \exp\left[ j\frac{2\pi}{\lambda} d_n \cos(\theta_i) + j\phi_n \right] \tag{7}$$

$fn(\theta i)$ is the radiation pattern of the $n$th cell in the direction $\theta i$.

A diagonal matrix **X** is added to control the weight of that differences in different angles. The weight matrix is optimized by DE subsequently.

$$[\mathbf{X}][\mathbf{A}][\mathbf{I}] = [\mathbf{X}][\mathbf{S}] \tag{8}$$

The left side of (8) represents the obtained pattern and the right side represents the desired pattern. In theory, (8) cannot be truly equal and there will always be some errors. After adding errors to both sides of the equation, using LSM, (8) becomes:

$$[\mathbf{XA} + \mathbf{E}][\mathbf{I}] = [\mathbf{XS}] + \varepsilon \tag{9}$$

**E** is an $M \times N$ dimensional correction matrix. $\varepsilon$ is the difference of the desired pattern and the obtained pattern. $[\varepsilon]_{1\times M} = [\varepsilon(\theta_1), \ldots, \varepsilon(\theta_i), \ldots, \varepsilon(\theta_M)]^T$.

In order to let (9) approximate the original equation (8) **E** and $\varepsilon$ should be as small as possible, that is $||[\mathbf{E}|\varepsilon]||_F = \min$.

(9) is equivalent to

$$\left( [\mathbf{XA}|\mathbf{XS}] + [\mathbf{E}|\varepsilon] \right) \begin{bmatrix} \mathbf{I} \\ -1 \end{bmatrix} = 0 \tag{10}$$

If $\mathbf{C} = [\mathbf{XA}|\mathbf{XS}], \Delta = [\mathbf{E}|\varepsilon], \mathbf{v} = \begin{bmatrix} \mathbf{I} \\ -1 \end{bmatrix}$, using LSM to find the nonzero solution **v** of following equation

$$(\mathbf{C} + \Delta)\mathbf{v} = 0 \tag{11}$$

The last component of **v** should not be zero. $\Delta$ should satisfy $||\Delta||_F = \mathbf{min}$.

The step of WTLSM is as follows:

Step 1. Calculate the initial diagonal matrix **X**:

$$\mathbf{X} = \begin{bmatrix} x_{11} & 0 & \cdots & 0 \\ 0 & x_{22} & \ddots & \vdots \\ \vdots & \ddots & \ddots & 0 \\ 0 & \cdots & 0 & x_{MM} \end{bmatrix} \tag{12}$$

$$x_{ii} = \frac{1}{S(\theta_i)}, (i = 1, 2, \ldots, M) \tag{13}$$

Step 2. The array manifold matrix **A** and the desired pattern vector **S** are combined.

$$\mathbf{C} = [\mathbf{XA} \mid \mathbf{XS}] \tag{14}$$

Step 3. Calculate the Hermitian conjugate matrix **CHC**. After the eigenvalue decomposition of matrix **CHC**, the eigenvectors vs. belonging to the minimal eigenvalue can be obtained.

$$\mathbf{V_s} = \begin{bmatrix} \mathbf{y} \\ t \end{bmatrix}, t \neq 0 \tag{15}$$

where $t$ is a constant.

Step 4. The solution solved by TLSM is as follows:

$$[\mathbf{I}]_{1\times N} = \frac{1}{t}[\mathbf{y}]_{1\times N} \tag{16}$$

### 2.2. Procedure of HDE-WTLSM

DE algorithm is simple and efficient. There are four steps in the algorithm: initialization, mutation, crossover and selection [10]. The algorithm steps have been depicted in detail in many literatures. Therefore, the description of them is not given in this paper.

The flow chat of HDE-WTLSM is given in Figure 1. In the whole algorithm process, the iteration procedure is only for the diagonal matrix **X**. The time consumption of our method is low and it is suited for large-scale array synthesis problems.

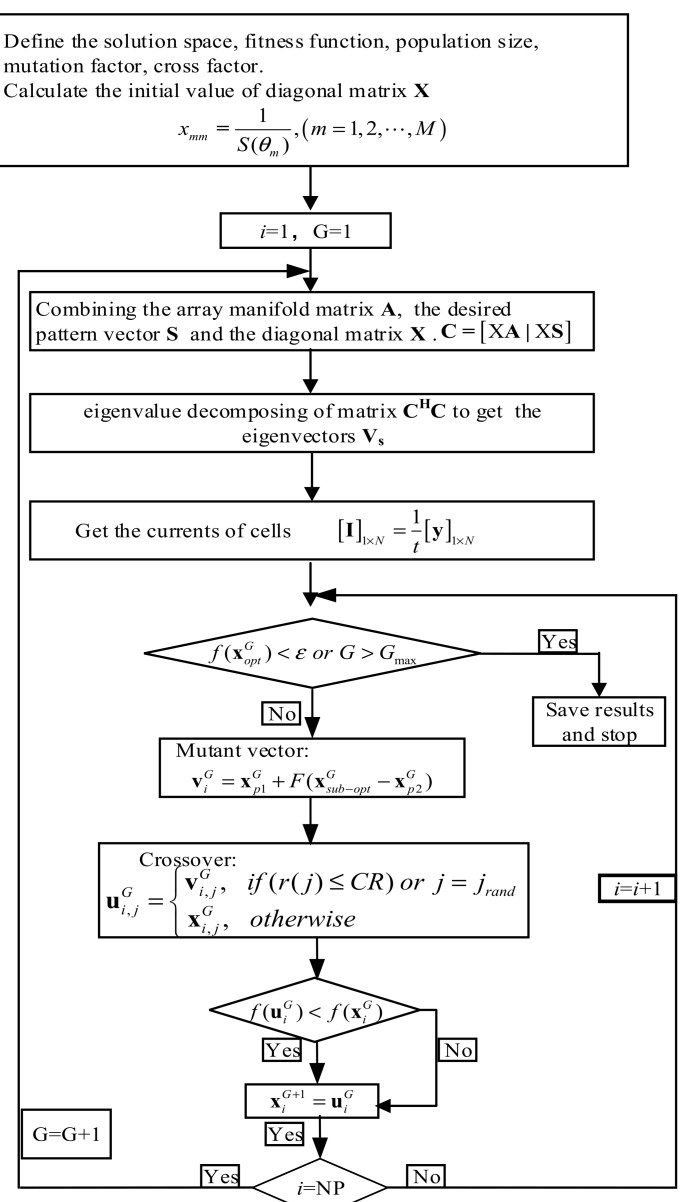

**Figure 1.** The flow chart of HDE-WTLSM.

### 3. Simulation Results

#### 3.1. Linear Array

To verify the proposed method, a 20-element uniform linear array with ideal point source is simulated. A low SLL and a null steering synthesis are required.

The fitness function is as follows:

$$fitness = U(F_0(\theta) - F_d(\theta))[\alpha\max|SLL - DSLL|$$
$$+\beta\max|NULL - DNULL|]$$

(17)

where $U(t) = \begin{cases} 1, t \geq 0 \\ 0, t < 0 \end{cases}$, $F_0(\theta)$ and $F_d(\theta)$ are the obtained and the desired array patterns, respectively. *SLL* denotes the obtained SLLs, *DSLL* is the desired SLLs, *NULL* denotes the depth of null depression optimized by the optimization algorithm. *DNULL* stands for the desired null depth. $\alpha$ and $\beta$ representative weights of SLL and null depth.

Since symmetrical arrays are used, only 10 elements' amplitudes need to be optimized. The excitation distribution is centrosymmetric with respect to the array.

### 3.1.1. Low SLL

The first example is the low SLL synthesis. In this example, the number of individuals in each population is 150, $F = 0.6$, cross factor $CR = 0.8$, $\alpha = 1, \beta = 0$. The desired SLL $DSLL = -50$ dB. The main beam-width is limited to $-12°\sim12°$.

After the optimization process, the normalized radiation pattern of low SLL synthesis is shown in Figure 2. In order to verify the performance of the algorithm, the results of two traditional numerical algorithms: Chebyshev and Taylor algorithm are also given in this figure. The comparison results of main lobe width and maximum SLL are given in Table 1. The corresponding convergence curves of DE and HDE-WTLSM are plotted in Figure 3. Figure 2 and Table 1 show that the SLL obtained by HDE-WTLSM algorithm is much lower than that of DE algorithm and it also has advantages over traditional numerical algorithm. Figure 3 shows that HDE-WTLSM has a faster convergence speed than DE.

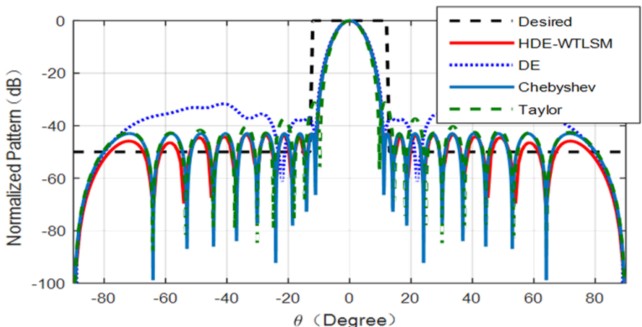

**Figure 2.** The normalized array patterns of low SLL synthesis.

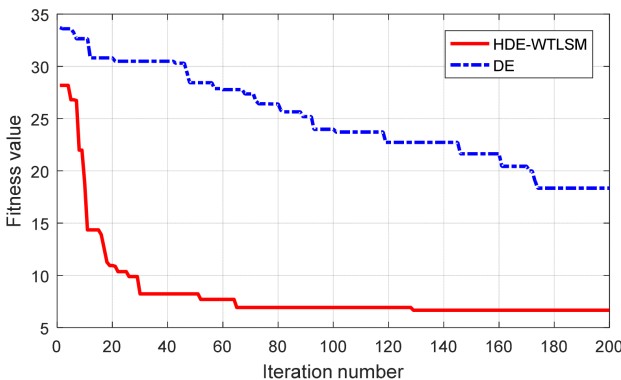

**Figure 3.** The convergent curves of low SLL synthesis.

**Table 1.** The comparison of results for low SLL synthesis.

| Algorithm | Main Lobe Width (°) | Maximum SLL (dB) |
|---|---|---|
| HDE-WTLSM | 21.88 | −43.22 |
| DE | 22 | −31.66 |
| Chebyshev | 22.24 | −43 |
| Taylor | 19.54 | −31.3 |

### 3.1.2. Null Steering

The second example is the null steering synthesis. In this example, the population number $NP = 150$, $F = 0.6$, $CR = 0.8$, $\alpha = 1$, $\beta = 2$, $DSLL = -50$ dB and $NULL = -80$ dB. The main beam-width is limited to $-15°{\sim}15°$. The null width is $10°$. The normalized array patterns of null steering synthesis are shown in Figure 4. The convergence curves of null steering synthesis are plotted in Figure 5. In this example, HDE-WTLSM also shows a better performance than DE. The results of example 1 and example 2 are obtained from Matlab.

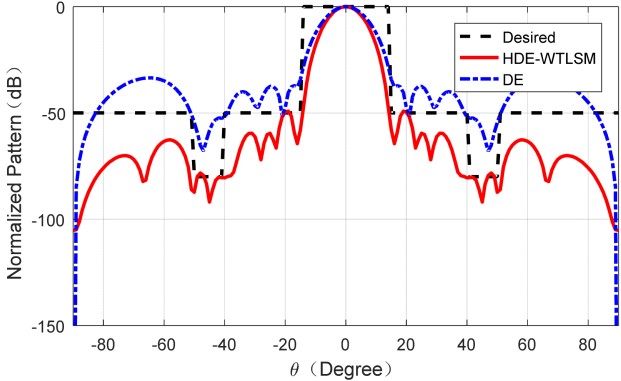

**Figure 4.** The convergent curves of null steering synthesis.

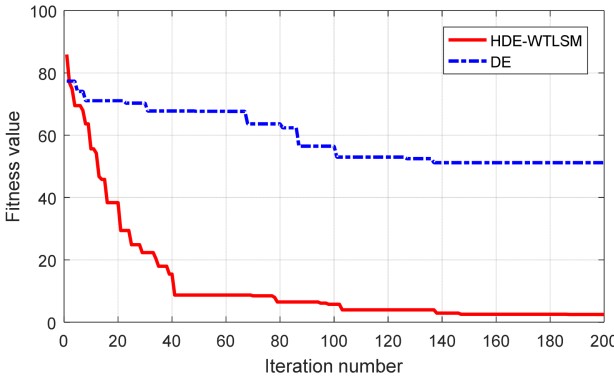

**Figure 5.** The convergent curves of null steering synthesis.

### 3.2. Conformal Truncated Conical Array

Circularly polarized antenna is conducive to the reception of space electromagnetic wave, so it is widely used in various communication systems [17–20]. To test the ability of our algorithm to optimize complex problems, a conformal truncated conical array with 39 right-hand circular polarization (RHCP) antennas is built. A single-port circular polarization element is used to reduce half of the unknowns. The structure of the antenna and the array is shown in Figure 6. The RHCP antenna is the same as reference [21]. The upper radius of the cone is 280 mm and the lower radius is 360 mm. The height of the cone is 400 mm. The central operating frequency is 1.575 GHz. The embedded patterns obtained by a high-frequency structure simulator (HFSS) are used in the algorithm to improve the accuracy of the synthesis. In this example, HDE-WTLSM is used to obtain low SLL synthesis and cross-polarization suppression simultaneously.

The fitness function is as follows:

$$
\begin{aligned}
fitness = U(F_0(\theta) - F_d(\theta))[\alpha \max|SLL - DSLL| \\
+ \beta \max|CROSS - DCROSS|]
\end{aligned}
\tag{18}
$$

where *CROSS* denotes the cross polarization level optimized by HDE-WTLSM. *DCROSS* stands for desired cross polarization level. $\alpha$ and $\beta$ representative weights of SLL and cross polarization level.

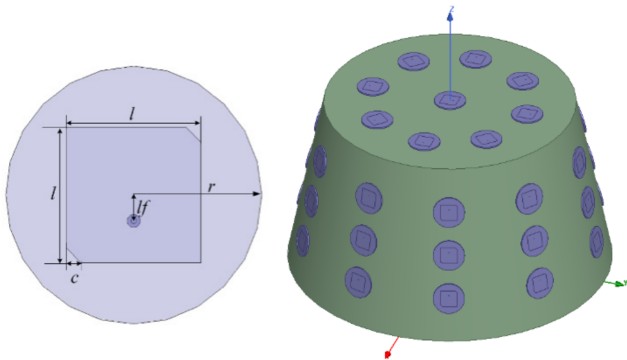

**Figure 6.** The RHCP antenna and the conformal truncated conical array.

In this example, the number of individuals in each population is 100, *F* = 0.6, *CR* = 0.8, $\alpha = 1, \beta = 1$. The desired SLL *DSLL* = −25 dB. The main beam-width is limited to −18°~18°. The desired cross polarization level *DCROSS* = −23 dB.

Unlike planar arrays, when the array is uniform feed, due to the influence of polarization, there is a depression in the zenith direction of the array radiation pattern. As shown in Figures 7 and 8. Due to the different orientations of each antenna in the conformal array, resulting in different angles between the co-polarization of each antenna. The results are the same for cross-polarization. This shows that polarization is the most difficult problem in conformal antenna array synthesis. If polarization is not considered in conformal antenna array synthesis, many unexpected results will be produced.

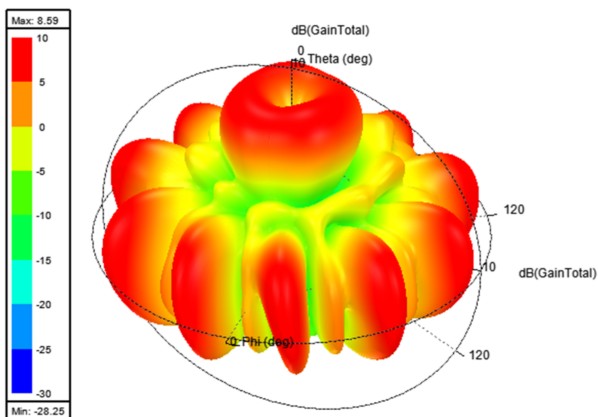

**Figure 7.** The 3D radiation pattern of uniform feed.

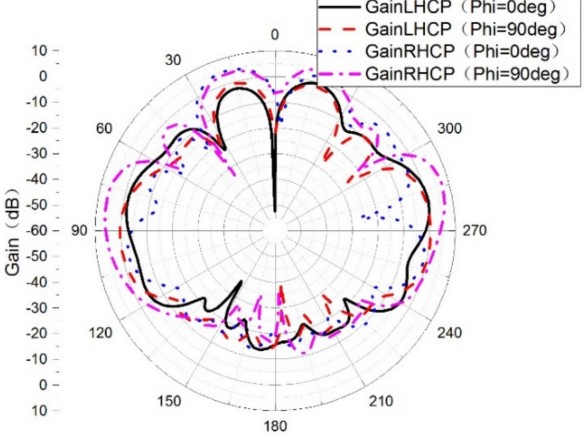

**Figure 8.** The 2D radiation pattern of uniform feed.

Then, HDE-WTLSM is used to reduce the effect of cross-polarization in this array. At the same time, it is hoped to get the result of low SLL in the main polarization direction. After synthesis, there is a main lobe in the zenith direction for co-polarization. In addition, the cross-polarization level decreased by about 20 dB. The results from Figures 7–10 were obtained from HFSS, which includes the active patterns of each antenna. These examples show that our algorithm can work in the array composed of ideal point sources and the actual conformal antenna array. The convergence curves of DE and HDE-WTLSM for the conformal array is given in Figure 11. As can be seen from the figure, the convergence speed of HDE-WTLSM algorithm is faster.

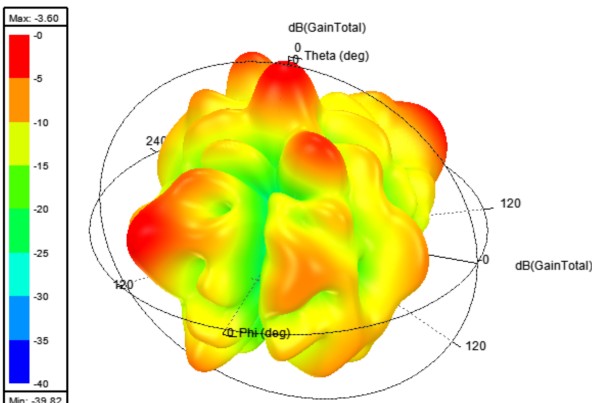

**Figure 9.** The 3D radiation pattern after synthesis.

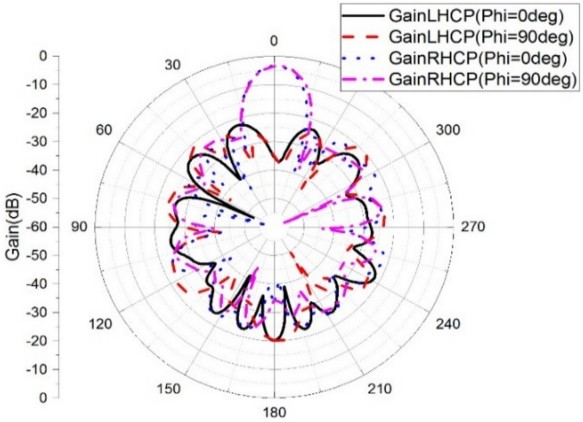

**Figure 10.** The 2D radiation pattern after synthesis.

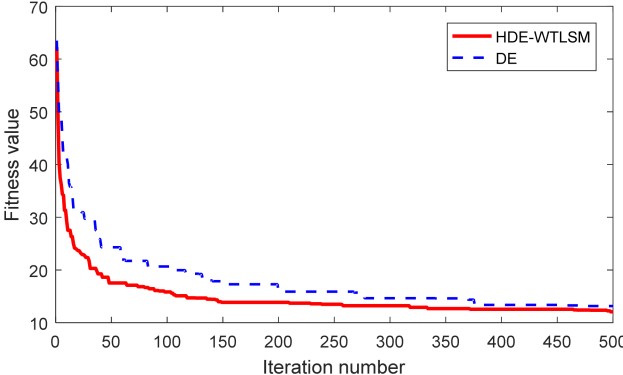

**Figure 11.** The convergent curves of low SLL synthesis for the conformal array.

## 4. Conclusions

In this paper, a novel algorithm named HDE-WTLSM has been proposed for antenna array pattern synthesis. Aweight matrix is added to TLSM to control the differences of desired and obtained pattern in different angles. DE is used to optimize that weight matrix. The algorithm is applied to low SLL and null steering pattern synthesis problems in a 20-elements linear array. Then, the algorithm is used to optimize a conformal truncated conical array to obtain low SLL and cross polarization level synthesis. Experiments show that HDE-WTLSM is very effective and has advantages of convergence speed.

**Author Contributions:** Conceptualization, R.L.; methodology, X.C.; software, J.W.; validation, Y.C.; formal analysis, L.X., investigation, Y.Y.; data curation, X.Y.; writing—original draft preparation, R.L.; writing—review and editing, L.X.; visualization, F.W.; funding acquisition, Y.Y. and X.Y. All authors have read and agreed to the published version of the manuscript.

**Funding:** This research was funded by Foundation of National Key Laboratory of Science and Technology on Space Microwave, grant number 2020SSFNKLSMT-08.

**Conflicts of Interest:** The authors declare no conflict of interest.

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
