# Peer review of "Array Pattern Synthesis Using a Hybrid Differential Evolution and Analytic Algorithm"

_electronics, doi:10.3390/electronics10182227_

Round 1

Reviewer 1 Report

This is an interesting topic to describe an optimization approach in antenna synthesis. However, in this manuscript, the state of the art is not clearly explained and the literature review is not very comprehensive. Some other comments for the authors to improve the work are as follows.

- The results are limited to simulation and the experimental part of the study is missing. 

- As the authors are proposing a new design method, it is crucial to evaluate the best approaches in the literature and compare the performance technically with the proposed one. 

- The other common applications for a similar algorithm in other applications should be specified.

- For all three design examples, it is suggested to design the antenna with 2 or 3 conventional methods and compare the achieved results with the proposed one preferably in a discussion section.

- Also, some minor punctuation errors could be seen in the manuscript that are needed to be corrected. For example, the spacing between Figs and numbers should be consistent.

Reviewer 2 Report

The manuscript titled “Array Pattern Synthesis Using a Hybrid Differential Evolution 2 and Analytic Algorithm” has certain new insights and mathematical analysis with simulation results. I recommend accepting the manuscript with minor revisions.

I suggest following corrections

  1. Add a performance comparison table with existing pattern synthesis algorithms (if possible add more references for existing literature comparison).
  2. The resolution of the figure 1 is too low and can be improved further.
  3. There are some sentences in the abstract with “we” and starting with “And”. These should be avoided.
  4. The author is advised to review and add the following suitable references.
  • LONG, S. A., SHEN, L. C., SCHAUBERT, D. H., FARRAR, F.G. “An experimental study of the circular-polarized elliptical printed circuit antenna. IEEE Transactions on Antennas and Propagation,1981, vol. 29, no. 1, p. 95–99. DOI: 10.1109/TAP.1981.1142549
  • SHARMA, P. C., GUPTA, K. C. Analysis and optimized design of single feed circularly polarized microstrip antennas. IEEE Transactions on Antennas and Propagation, 1983, vol. 31, no. 6, p. 949–955. DOI: 10.1109/TAP.1983.1143162
  • Darimireddy, N. K., Reddy, R. R., & Prasad, A. M. (2018). Asymmetric and symmetric modified bow‐tie slotted circular patch antennas for circular polarization. ETRI Journal40(5), 561-569.
  • Darimireddy, Naresh K., R. Ramana Reddy, and A. Mallikarjuna Prasad. "Asymmetric Triangular Semi-Elliptic Slotted Patch Antennas for Wireless Applications." Radioengineering 27, no. 1 (2018): 85.
  1. There are numerous punctuation and grammatical errors in the manuscript. Proofread for the whole document is required and necessary.
